# Feeding Behavior, Growth Performance and Meat Quality Profile in Broiler Chickens Fed Multiple Levels of Xylooligosaccharides

**DOI:** 10.3390/ani13162582

**Published:** 2023-08-10

**Authors:** Xixi Li, Xiaohong Wu, Wenfeng Ma, Houqiang Xu, Wei Chen, Furong Zhao

**Affiliations:** 1College of Animal Science and Technology, Henan University of Science and Technology, Luoyang 471023, China; lixixi0611@163.com (X.L.); lemonwxh@sina.com (X.W.); a113boy@163.com (W.M.); 2Key Laboratory of Animal Genetics, Breeding and Reproduction in the Plateau Mountainous Region, Guizhou University, Guiyang 550025, China; gzdxxhq@163.com (H.X.); chenweigzu@163.com (W.C.)

**Keywords:** xylooligosaccharides, behavior, performance indicators, meat characteristics, broiler

## Abstract

**Simple Summary:**

In recent years, after the use of antibiotic growth promoters was banned, scientists began to pay attention to new feed additives that can replace the use of antibiotics to improve the growth performance of animals and maintain good health. However, it is unclear whether feeding broiler chickens with different levels of xylooligosaccharides (XOS) affects the feeding behavior, thereby affecting broiler growth performance, and whether the addition of XOS has an effect on meat quality. Therefore, this study aimed to investigate the effects of XOS supplementation on feeding behavior, growth performance, slaughter performance, and meat quality in broiler chickens.

**Abstract:**

A total of 240 1-day-old Arbor Acres broiler chickens were randomly distributed to 4 treatment groups with 6 replicates and 10 birds per replicate. Chickens were fed with corn-soybean meal diet supplementation with additions of 0, 150, 300, and 450 mg/kg XOS for 42 days. At 4 weeks of age, the average feeding time was reduced in the 450 mg/kg XOS group (*p* < 0.05), and the percentage of feeding time was increased in the 300 mg/kg XOS group (*p* < 0.05). At 5 weeks of age, broilers fed with 300 mg/kg XOS had increased the percentage of feeding time (*p* < 0.05), and 450 mg/kg XOS had increased the feeding frequency and percentage of feeding time (*p* < 0.05). At 6 weeks of age, the feeding frequency was highest in the 450 mg/kg XOS group (*p* < 0.05). During 4 to 6 weeks of age, the average feeding time was increased in 300 mg/kg XOS group (*p* < 0.05), the frequency was improved in the 450 mg/kg XOS group (*p* < 0.05), and the percentage of feeding time was longer in the XOS group than that in the control group (*p* < 0.05). The average daily gain was improved during days 22–42 and days 1–42 in the 150 mg/kg XOS group (*p* < 0.05). Broilers fed with 300 mg/kg XOS had an increased eviscerated rate (*p* < 0.05). The pH_45min_ of breast muscle was highest in the 450 mg/kg XOS group (*p* < 0.05), as well as the pH_45min_ and pH_24h_ of thigh muscle, which improved in the 300 mg/kg and 450 mg/kg XOS groups (*p* < 0.05). In addition, the cooking loss of thigh muscle was reduced in the 300 mg/kg XOS group (*p* < 0.05). In conclusion, dietary supplementation with XOS had positive effects on the feeding behavior, growth performance, and meat quality of broiler chickens.

## 1. Introduction

Since 1986, Sweden, the European Union, the United States, Japan, South Korea and many other countries have banned the addition of growth-promoting antibiotics in feed in livestock and poultry breeding, and China has implemented a total ban on the use of growth-promoting antibiotics in July 2020. Due to the development of antimicrobial resistance brought on by the use of antibiotics as feed additives, there is a higher risk of morbidity and death from diseases that were previously curable with antibiotics [1]. In view of these, considerable attention has been focused on the development for substitutes to antibiotics to maintain poultry production performance, such as prebiotics, plant extracts, bioactive medicinal herbs, acidifiers, amino acids, and probiotics [2,3,4,5,6,7].

Prebiotics are food components that specifically promote the activity and growth of particular species of gut bacteria, typically bifidobacteria and lactobacilli, with positive health effects [8,9]. All prebiotics have been described as short-chain carbohydrates with a degree of polymerization between two and sixty, and it is believed that neither human nor animal digestive enzymes can break them down [10].

Xylooligosaccharides (XOS) are oligomers that show a broad degree of polymerization and are composed of xylose units linked by β, 1-4 glycosidic bonds [11]. They are mainly composed of xylobiose, xylotriose, xylotetraose, and small amounts of xylopenose and xylohexose [12]. Generally, xylan-rich lignocellulosic materials (LCMs) such as corn cobs, olives, rice husks, and cottonseed husks are directly hydrolyzed to produce XOS [13]. Given that antibiotics were banned from animal feed, XOS may be a potential alternative to antibiotics due to their function in protecting the gastrointestinal tract from pathogenic microbial communities, thus maintaining productivity and improving the quality of livestock products [14]. 

XOS has prebiotic properties such as improving growth rate; regulating lipid metabolism; modulating nutrient digestibility; and exerting antioxidant effects [15,16,17,18]. Reported that dietary supplementation of 150 mg/kg XOS improved the laying rate, daily egg production, yolk ratio and decreased the feed gain ratio of laying hens in the late laying period [19]. Administration of dietary XOS increased the average daily gain and improved the feed efficiency of broilers by reducing the feed intake [20,21,22]. At the same time, the addition of synbiotic consisting of prebiotics (XOS and yeast wall) and probitics to the diet of Qingjiao Ma chickens was beneficial to improve meat quality by reducing the drip loss and cooking loss of breast muscle after slaughter [23]. 

Feeding behavior is a fundamental activity of all animals that can be modulated by internal energy states or external sensory signals, plays a decisive role in animal growth and reproduction, and promotes the expression of biological functions that are meaningful to animal welfare. However, no study has reported the effects of XOS on the feeding behavior of broilers. Therefore, the objective of this study was to evaluate the effects of dietary supplementation with XOS on the feeding behavior, growth performance, and meat quality of broilers.

## 2. Materials and Methods

All the animal experiments were approved by the Institutional Animal Care and Use Committee of the Henan University of Science and Technology (HAUSTEAW-2021-C00227).

### 2.1. Feeding Management and Experience Design

Broilers were raised in wire cages with a length, width, and height of 95 cm × 90 cm × 40 cm and allowed to feed and drink ad libitum. The room temperature was maintained at 33–35 °C for the first week and gradually decreased by 2–3 °C every week until the room temperature reached 25 °C. During the trial period, a lighting program of 16 h light and 8 h of darkness was adopted.

A total of 240 1-day-old Arbor Acres broilers were randomly distributed to 4 treatment groups (6 replicates per treatment with 10 birds per replicate). Broilers were fed a corn-soybean meal basal diet supplemented with 0 (control), 150, 300, and 450 mg/kg XOS. The experimental trial period lasted for 42 days and was divided into two stages: starter (1 to 21 days) and grower (22 to 42 days). The basal diet were formulated to meet the nutritional requirements of the National Research Council [24] (Table 1). XOS was purchased from Henan Keenbo Agricultural Technology Co., Ltd. (Henan, China). The main components of XOS were xylobiose, xylotriose, and xylotetraose, and the rest were cellulose and hemicellulose, in which the content of XOS was 20%.

### 2.2. Measurement of Feeding Behavior

The feeding behavior indicators refer to the methods [25]. During the experiment, 8 birds were randomly selected as target chickens for each treatment, and different parts of the chickens were marked with biological dyes. As the chickens exhibited some adaptability to the surrounding environment and were relatively stable, we used the Xiaoyi smart camera (night vision version) to shoot continuously for 24 h at the same time every week, starting at 4 weeks of age. The Observer XT 12 behavior analysis software (Noldus Information Technology Co., Ltd., Wageningen, The Netherlands) was used for analysis after the video was converted on the computer, and the maximum feeding time, minimum feeding time, average feeding time, feeding frequency, and percentage of feeding time in each treatment were manually recorded. The maximum, minimum, and average feeding times indicate the maximum, minimum, and average time of continuous feeding in a unit time (1 h). The feeding frequency is the number of feeding behaviors that occurred in a unit time (1 h). The percentage of feeding time is the percentage (%) of the duration of feeding behavior in the total observation time (24 h).

### 2.3. Growth Performance

The weight and feed consumption of broilers were recorded every week with replicate as the unit. Average daily gain (ADG), average daily feed intake (ADFI), and feed/gain ratio (F/G) were calculated from 1 to 21 days (starter), 22 to 42 days (grower), and 1 to 42 days (the whole period).

### 2.4. Slaughter Performance

At 42 days of age, two birds with close to average body weight were selected from each replicate and dissected to determine slaughter performance, including slaughter rate, eviscerated rate, breast muscle rate, thigh muscle rate, and abdominal fat rate.

### 2.5. Meat Quality

The lightness (L*), redness (a*), and yellowness (b*) of three positions at the thickest part of the right pectoral breast and thigh muscle were measured by an automatic colorimeter (NR60CP, ThreeNH Technology Co., Ltd., Shenzhen, China), and the average value was taken. A portable pH meter (HI-99163, Beijing Hanna Instruments Science and Technology Co., Ltd., Beijing, China) was used to measure the pH of three positions of the right pectoral breast and thigh muscle at 45 min and 24 h after slaughter, and the average value was taken. A meat sample with a length, width, and height of 3 cm × 1 cm × 1 cm was taken from the left pectoral breast muscle and thigh muscle. The weight was recorded as m1, and the meat sample was hung in a plastic bag with a thin wire. The sample should not touch the plastic bag. It was placed in a 4 °C refrigerator and taken out after 24 h. A filter paper was used to dry the surface moisture of the meat sample, and the weight was recorded as m_2_. Furthermore, the drip loss was calculated.
(1)$$Drip\;loss\;(\%)=100\times (m_{1}-m_{2})/m_{1}$$

A meat sample with a length, width, and height of 3 cm × 1 cm × 1 cm was taken from the left pectoral breast muscle and thigh muscle after slaughter, and the weight was recorded as m_1_. The sample was immersed in a water bath at 80 °C and heated until the center temperature of the meat reached 75 °C. Furthermore, it was cooled to room temperature. A filter paper was used to absorb the surface moisture, and the weight was recorded as m_2_. Furthermore, the cooking loss was calculated.
(2)$$Cooking\;loss\;(\%)=100\times (m_{1}-m_{2})/m_{1}$$

The shear force of the meat samples after cooking loss was measured with a muscle tenderness meter (C-LM3B tenderization instruments, Northeast Agricultural University, Harbin, China). During the measurement, the direction of the muscle fiber was perpendicular to the incision. Each meat sample was measured three times, and the average value was taken.

### 2.6. Statistical Analysis

SPSS 20.0 software (IBM Inc., Armonk, NY, USA) was used for statistical analysis. All data were analyzed by the polynomial contrasts test, linear, and cubic. The analysis results were expressed as mean values and standard errors, and the significance level was set at *p* < 0.05.

## 3. Results

### 3.1. Feeding Behavior

The results of feeding behavior are shown in Table 2, Table 3, Table 4 and Table 5. At 4 weeks of age, dietary supplementation of 450 mg/kg XOS significantly reduced the average feeding time of broilers compared with the control, 150, and 300 mg/kg XOS groups (*p* < 0.05), and the percentage of feeding time showed higher values in the 300 mg/kg XOS group than those in the control, 150, and 450 mg/kg XOS groups (*p* < 0.05). At 5 weeks of age, the average feeding time was affected quadratically (*p* = 0.015). The feeding frequency in the 450 mg/kg XOS group was higher than that in other treatment groups (*p* < 0.05). In addition, the percentage of feeding time significantly increased in the 300 and 450 mg/kg XOS groups compared with the control group (*p* < 0.05), and the 450 mg/kg XOS group was higher than that in the 150 mg/kg XOS group (*p* < 0.05). At 6 weeks of age, the broilers in the 450 mg/kg XOS group had a significantly higher feeding frequency than those in the control and 300 mg/kg XOS groups (*p* < 0.05). During 4 to 6 weeks of age, the average feeding time was substantially longer in the 300 mg/kg XOS group than that in the control and 450 mg/kg XOS groups (*p* < 0.05), and the 450 mg/kg XOS group had the highest feeding frequency among all groups (*p* < 0.05), and the percentage of feeding time in the XOS groups had significantly increased compared with that in the control group (*p* < 0.05). In each week, there were no appreciable differences between the groups in the maximum time and minimum time (*p* > 0.05).

### 3.2. Growth Performance

Dietary supplementation with XOS exhibited a positive effect on growth performance in broilers (Table 6). Although there were no significant differences in ADFI and F/G among all groups (*p* > 0.05), the ADG of broilers significantly increased in the 150 mg/kg XOS group compared with control group during 22–42 days and 1–42 days (*p* < 0.05).

### 3.3. Slaughter Performance

As shown in Table 7, broilers supplemented with 300 mg/kg XOS showed a higher eviscerated rate compared with the control, 150 mg/kg, and 450 mg/kg XOS groups (*p* < 0.05). However, there was no influence on slaughter rate, breast muscle, thigh muscle, or abdominal fat among all treatment groups (*p* > 0.05).

### 3.4. Meat Quality

As shown in Table 8, Dietary supplementation of 450 mg/kg XOS had improved the pH_45min_ for breast muscle compared with the control and 150 mg/kg XOS groups (*p* < 0.05); the cooking loss and shear force were affected linearly (*p* = 0.043 and *p* = 0.020, respectively) and decreased in a dose-dependent manner. The supplementation of 300 mg/kg and 450 mg/kg XOS had improved pH_45min_ and pH_24h_ for thigh muscle compared with the control group (*p* < 0.05). The cooking loss was significantly reduced in the 300 mg/kg XOS group than in the control group (*p* < 0.05). There was no significant difference between treatment groups in other indicators (*p* > 0.05).

## 4. Discussion

Chickens have the instinct to feed for energy, and feeding behavior is dominated by alternating hunger and satiety. Hunger and satiety are the most important factors affecting animal feeding behavior; the alternate domination of hunger and satiety is the internal mechanism regulating the feeding behavior of animals [26,27]. Feeding behavior directly determines the feed-to-meat ratio, which in turn affects the economic value of poultry. During 4 to 6 weeks of age, although the results of this study showed that different levels of XOS had no effect on maximum and minimum feeding time, the percentage of feeding time of broiler chickens increased. Furthermore, dietary supplementation of 300 mg/kg XOS had a longer average feeding time, and 450 mg/kg XOS had an improved feeding frequency in broiler chickens. The increase in feeding frequency and percentage of feeding time were conducive to promoting the feeding activities of broilers [25]. A higher and longer satiation reduces the motivation to feed for a longer time and increases the intervals between meals [28]. On the contrary, the sense of hunger will stimulate feeding behavior and improve feeding frequency. This may be related to the function of XOS in promoting intestinal peristalsis and accelerating the intestinal emptying rate [29]. The effect of promoting intestinal tract movement was strengthened with the increase in XOS supplemental levels, and feed was excreted before it was fully digested and absorbed. Therefore, broilers could meet their food requirements by increasing feeding time and frequency. However, the mechanisms of feeding behavior remain unclear and need to be further studied in the future.

Several studies have reported that dietary XOS has a beneficial effect on the growth performance of broilers. The addition of 0.5% AXOS to the wheat-based diet and 0.25% AXOS to the maize-based diet increased the FCR, thereby improving the nutrient utilization of broilers [30]. The coccidium-infected broilers fed 0.5 g/kg XOS alleviated negative effects on growth performance and nutrient utilization [31]. It has been demonstrated that dietary supplementation of XOS in broilers exerted an “overdosing” effect, where a low incorporation rate of 0.1 g/kg had a beneficial effect on broiler body weight, whereas a high incorporation rate had negligible or no effect on performance [32], which was in agreement with our findings. However, no significant influence was observed on ADG, final body weight gain, ADFI, and FCR of broilers when provided 0, 50, and 100 g/t XOS to corn-soybean meal diets [33]. The present study found that 150 mg/kg XOS significantly increased ADG in broilers aged 22–42 days and 1–42 days, in contrast with the control group. Feeding behavior is related to resting behavior and, therefore, growth performance [34]. Research shows that there was a significant negative correlation between the daily feeding frequency and the ADG of chickens, and breeding chickens with a lower feeding frequency could improve production performance [35]. The increased feeding frequency in broilers fed with 450 mg/kg XOS in the grower stage means that more energy is consumed along with feeding activities, and energy utilization efficiency decreased, which is not profit to weight gain. The positive growth-promoting effect may be attributed to XOS stimulating the proliferation of beneficial bacteria such as bifidobacteria and lactobacilli [21]. The villus height of duodenum, jejunum, and ileum of broilers was longer through adding 150 mg/kg XOS [36], and the increased number of short-chain fatty acids (SCFAs) secreted by beneficial bacteria could reduce the pH of the intestine, provide a suitable working environment for digestive enzymes, and promote the absorption and utilization of nutrients [37,38,39]. XOS can enhance the expression of genes involved in carbohydrate transport metabolism, such as ABC transporter, amylase, xylosidase, galactosidase, and glucosidase [40]. The percentage of feeding time had increased during 22–42 days, while the supplementation of 300 and 450 mg/kg XOS had no effect on growth performance. This is possibly due to the overacidic intestinal environment, which affects the activity of digestive enzymes, or XOS, which promotes the rate of emptying the digestive tract and adversely affects the absorption of nutrients. The effect of XOS on the grower stage is better than that of the starter stage, which may be due to the fact that the digestive tract system of birds in the starter stage is not fully developed, thus causing intestinal discomfort in young animals. Therefore, XOS should be moderately added to the actual production.

The positive effect was observed when broilers supplemented with 200 mg/kg XOS improved the percentages of breast and thigh muscle [41]. However, our results showed that the slaughter rate, breast muscle rate, thigh muscle rate, and abdominal fat rate did not change by adding XOS to the broiler diet, while 300 mg/kg XOS significantly increased the eviscerated rate. Similar results reported that dietary XOS had no significant effect on the breast muscle rate and thigh muscle rate of broilers [42].

Meat color, pH, shear force, drip loss, and cooking loss are important indicators that reflect meat quality [43,44]. Meat color is the most direct sensory characteristic index to evaluate the appearance of meat, and L*, a*, and b* are used as the basis for evaluation [45]. Muscle pH at 45 min after slaughter was used to judge meat quality and its classification into normal, PSE (pale, soft, exudative), and DFD (dark, firm, dry) types [46]. After the animals are slaughtered, the supply of nutrients to their circulation is interrupted, causing a glycolytic response that produces lactic acid and lowers the pH in the muscle. A previous study showed that the application of XOS in growing-finishing pigs can decrease the drip loss of dorsal longissimus muscle, indicating that XOS has the effect of improving muscle water-holding capacity, and this effect was gradually strengthened with the increase in XOS supplementation [47]. There was no influence of XOS supplementation on meat color or pH, but the drip loss of thigh muscle in broilers linearly decreased with the increasing XOS supplementation level [48]. However, in our experiment, dietary supplementation of 450 mg/kg XOS had increased pH_45min_ of breast muscle, and the addition of 300 and 450 mg/kg XOS had increased pH_45min_ and pH_24h_ of thigh muscle. In addition, broilers fed with 300 mg/kg XOS reduced the cooking loss of the thigh muscle. The accumulation of lactic acid in the tissues decreases meat pH, resulting in less water that protein can attract and retain [49]. Research shows that XOS promotes the absorption of minerals by the body [50]. When the content of Mg^2+^ in blood increases, it is beneficial to increase the pH value of muscle, reduce the speed of glycolysis, and thus maintain the pH value of muscle at a high level. Furthermore, XOS has good antioxidant properties [44,51], including the prominent 2,2′-dipheny-1-picryl-hydrazyl, 2,2′-azino-bis (3-ethylbenzothiazoline-6-sulfonic acid), and iron-reducing antioxidant capacity [52]. XOS, as an antioxidant, can scavenge free radicals produced by cell metabolism and alleviate the lipid peroxidation of the cell membrane [53]; this may be the reason for the decreased cooking loss of the muscle.

## 5. Conclusions

In conclusion, this study confirmed that XOS can promote feeding behavior and produce positive effects on growth performance through increasing ADG during the grower stage. At the same time, dietary supplementation with XOS can also improve the pH value and reduce the cooking loss of meat after slaughter.

## Figures and Tables

**Table 1 animals-13-02582-t001:** Composition and nutrient levels of the basal diets (air-dry basis).

Item	1–21 Days	22–42 Days
Ingredients, %		
Corn	60.80	62.55
Soybean meal	30.00	27.50
Corn gluten meal	3.00	2.00
Soybean oil	2.00	4.00
Limestone	1.60	1.60
CaHPO_4_	1.30	1.30
NaCl	0.30	0.30
Lysine	0.35	0.20
Methionine	0.20	0.10
Mildew preventive	0.08	0.08
Antioxidant	0.06	0.06
Choline	0.08	0.08
Premix ^1^	0.23	0.23
Total	100.00	100.00
Nutrient, % ^2^		
ME (MJ·kg^−1^)	12.55	13.13
Crude protein	21.00	19.00
Ether extract	4.50	6.95
Crude fiber	2.20	2.20
Crude ash	5.60	4.49
Calcium	1.02	1.00
Total phosphorus	0.55	0.53
Digestible tryptophan	0.24	0.22
Digestible methionine	0.52	0.39
Digestible Lysine	1.27	1.08
Digestible threonine	0.78	0.72

^1^ Premix provided the following per kg of the diet: vitamin A, 6500 IU; vitamin D3, 3000 IU; vitamin E, 80 IU; vitamin K, 5 mg; vitamin B1, 4 mg; vitamin B2, 5.5 mg; vitamin B6, 5 mg; niacin, 30 mg; pantothenic acid, 12 mg; folic acid, 1 mg; Mn, 80 mg; Fe, 110 mg; Cu, 12 mg; Zn, 70 mg; I, 0.4 mg; Se, 0.2 mg. ^2^ Nutrient levels are calculated values.

**Table 2 animals-13-02582-t002:** Effects of XOS on feeding behavior of broilers at 4 week of age.

Parameters	Dietary XOS Levels (mg/kg)				SEM	*p*-Values		
	0	150	300	450		Anova	L	Q
Maximum feeding time (s)	166.93	180.14	184.76	156.12	13.50	0.071	0.540	0.086
Minimum feeding time (s)	8.39	8.56	15.92	13.64	4.01	0.203	0.071	0.181
Average feeding time (s)	54.36 ^a^	57.49 ^a^	63.25 ^a^	42.40 ^b^	5.21	0.009	0.132	0.005
Feeding frequency (number)	5.26	6.38	7.30	7.49	0.873	0.084	0.009	0.025
Percentage of feeding time (%)	9.21 ^b^	10.74 ^b^	13.72 ^a^	8.69 ^b^	1.24	0.007	0.776	0.006

SEM: Standard error means; L: Linear effect; Q: Quadratic effect; ^a,b^: In the same row, values with the different letter superscripts means significantly difference (*p* < 0.05).

**Table 3 animals-13-02582-t003:** Effects of XOS on feeding behavior of broilers at 5 week of age.

Parameters	Dietary XOS Levels (mg/kg)				SEM	*p*-Values		
	0	150	300	450		Anova	L	Q
Maximum feeding time (s)	171.39	158.49	185.92	167.14	14.42	0.186	0.756	0.918
Minimum feeding time (s)	8.11	7.64	8.29	5.61	1.90	0.359	0.255	0.376
Average feeding time (s)	46.90 ^b^	56.16 ^ab^	60.84 ^a^	50.65 ^ab^	4.73	0.063	0.345	0.015
Feeding frequency (number)	6.45 ^b^	5.91 ^b^	6.48 ^b^	8.26 ^a^	0.536	0.006	0.003	0.000
Percentage of feeding time (%)	8.26 ^c^	9.30 ^bc^	10.83 ^ab^	11.50 ^a^	0.968	0.010	0.001	0.003

SEM: Standard error means; L: Linear effect; Q: Quadratic effect; ^a,b,c^: In the same row, values with the different letter superscripts means significantly difference (*p* < 0.05).

**Table 4 animals-13-02582-t004:** Effects of XOS on feeding behavior of broilers at 6 week of age.

Parameters	Dietary XOS Levels (mg/kg)				SEM	*p*-Values		
	0	150	300	450		Anova	L	Q
Maximum feeding time (s)	172.54	195.02	163.78	167.46	28.37	0.828	0.603	0.785
Minimum feeding time (s)	8.50	15.67	9.41	11.63	4.78	0.953	0.840	0.758
Average feeding time (s)	49.06	53.48	51.95	45.52	5.76	0.536	0.508	0.331
Feeding frequency (number)	6.25 ^bc^	7.39 ^ab^	5.83 ^c^	7.70 ^a^	0.599	0.017	0.207	0.346
Percentage of feeding time (%)	8.13	10.84	8.26	9.68	1.10	0.118	0.592	0.658

SEM: Standard error means; L: Linear effect; Q: Quadratic effect; ^a,b,c^: In the same row, values with the different letter superscripts means significantly difference (*p* < 0.05).

**Table 5 animals-13-02582-t005:** Effects of XOS on feeding behavior of broilers during 4 to 6 weeks of age.

Parameters	Dietary XOS Levels (mg/kg)				SEM	*p*-Values		
	0	150	300	450		Anova	L	Q
Maximum feeding time (s)	170.29	177.88	178.15	163.57	12.33	0.716	0.612	0.391
Minimum feeding time (s)	8.33	10.62	11.32	10.32	2.03	0.339	0.303	0.304
Average feeding time (s)	50.11 ^bc^	55.71 ^ab^	58.68 ^a^	46.19 ^c^	2.80	0.004	0.452	0.000
Feeding frequency (number)	5.99 ^b^	6.56 ^b^	6.54 ^b^	7.82 ^a^	0.443	0.009	0.001	0.002
Percentage of feeding time (%)	8.53 ^b^	10.29 ^a^	10.94 ^a^	9.96 ^a^	0.610	0.014	0.031	0.001

SEM: Standard error means; L: Linear effect; Q: Quadratic effect; ^a,b,c^: In the same row, values with the different letter superscripts means significantly difference (*p* < 0.05).

**Table 6 animals-13-02582-t006:** Effects of XOS on growth performance of broilers.

Parameters	Dietary XOS Levels (mg/kg)				SEM	*p*-Values		
	0	150	300	450		Anova	L	Q
Day 1 to 21								
ADG (g/d)	15.56	17.13	15.98	15.76	1.09	0.164	0.879	0.513
ADFI (g/d)	30.57	31.56	30.73	30.37	1.89	0.535	0.807	0.849
F/G	1.98	1.85	1.93	1.94	0.106	0.241	0.905	0.667
Day 22 to 42								
ADG (g/d)	69.32 ^b^	76.59 ^a^	71.75 ^ab^	72.27 ^ab^	3.15	0.032	0.711	0.346
ADFI (g/d)	120.36	119.55	117.24	119.89	6.02	0.610	0.840	0.900
F/G	1.74	1.56	1.64	1.67	0.08	0.241	0.638	0.212
Day 1 to 42								
ADG (g/d)	42.44 ^b^	46.86 ^a^	43.86 ^ab^	44.02 ^ab^	1.76	0.021	0.779	0.280
ADFI (g/d)	75.46	75.56	73.99	75.13	3.59	0.667	0.815	0.952
F/G	1.78	1.61	1.69	1.71	0.075	0.056	0.648	0.210

SEM: Standard error means; L: Linear effect; Q: Quadratic effect; ^a,b^: In the same row, values with the different letter superscripts means significantly difference (*p* < 0.05); ADG: average daily gain; ADFI: average daily feed intake; F/G: feed to gain ratio.

**Table 7 animals-13-02582-t007:** Effects of XOS on slaughter performance of broilers (% of live body weight).

Parameters	Dietary XOS Levels (mg/kg)				SEM	*p*-Values		
	0	150	300	450		Anova	L	Q
Slaughter rate	91.41	92.34	91.81	92.05	0.441	0.064	0.344	0.367
Eviscerated rate	76.24 ^b^	76.52 ^b^	79.40 ^a^	77.36 ^b^	0.975	0.004	0.090	0.082
Breast muscle rate	24.43	25.39	25.92	25.95	0.944	0.123	0.091	0.191
Thigh muscle rate	23.08	23.27	22.81	22.11	1.17	0.335	0.353	0.566
Abdominal fat rate	2.12	1.84	2.11	2.02	0.700	0.313	0.964	0.877

SEM: Standard error means; L: Linear effect; Q: Quadratic effect; ^a,b^: In the same row, values with the different letter superscripts means significantly difference (*p* < 0.05).

**Table 8 animals-13-02582-t008:** Effects of XOS on meat quality of broilers.

Parameters	Dietary XOS Levels (mg/kg)				SEM	*p*-Values		
	0	150	300	450		Anova	L	Q
Breast muscle								
L*	43.65	42.25	42.51	41.95	1.17	0.162	0.192	0.383
a*	1.70	1.82	1.62	2.20	0.64	0.375	0.508	0.708
b*	5.37	5.81	5.28	6.31	0.796	0.211	0.362	0.583
pH_45min_	6.34 ^bc^	6.23 ^c^	6.57 ^ab^	6.66 ^a^	0.119	0.002	0.003	0.008
pH_24h_	6.06	5.96	6.02	6.00	0.064	0.132	0.505	0.531
Drip loss (%)	3.31	2.99	3.31	2.85	0.398	0.261	0.400	0.688
Cooking loss (%)	28.82 ^a^	28.56 ^a^	26.76 ^ab^	24.74 ^b^	2.15	0.072	0.043	0.113
Shear force (N)	33.29 ^a^	32.59 ^a^	30.07 ^ab^	29.22 ^b^	1.94	0.097	0.020	0.072
Thigh muscle								
L*	45.15	44.68	42.77	44.12	1.44	0.115	0.285	0.387
a*	2.58	2.02	2.47	2.51	0.653	0.397	0.909	0.800
b*	6.96	5.63	6.23	5.88	0.993	0.193	0.398	0.552
pH_45min_	6.38 ^b^	6.53 ^ab^	6.72 ^a^	6.74 ^a^	0.100	0.002	0.001	0.002
pH_24h_	6.31 ^b^	6.43 ^ab^	6.58 ^a^	6.49 ^a^	0.082	0.023	0.027	0.015
Drip loss (%)	2.94	2.64	3.75	2.80	0.688	0.124	0.753	0.775
Cooking loss (%)	29.64 ^a^	26.37 ^ab^	21.86 ^b^	25.92 ^ab^	2.408	0.004	0.073	0.027
Shear force (N)	31.60	34.46	32.12	32.57	2.72	0.305	0.948	0.820

SEM: Standard error means; L: Linear effect; Q: Quadratic effect; ^a,b,c^: In the same row, values with the different letter superscripts means significantly difference (*p* < 0.05).

## Data Availability

Not applicable.

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
