# Peer review of "Feeding Behavior, Growth Performance and Meat Quality Profile in Broiler Chickens Fed Multiple Levels of Xylooligosaccharides"

_animals, 2023, doi:10.3390/ani13162582_

Round 1

Reviewer 1 Report

I read this manuscript for possible publication in Animals. This paper deals with very interesting topic. This review manuscript is written correctly and for the benefit of the reader. The authors of this paper have taken a very important and topical issue which is the feeding behavior, growth performance and meat quality in broiler chickens. The manuscript is quite clear. The introduction is practically comprehensive, concise, well written, and contains the needed information. The studies were carried out in according to well-proven methods. Results, discussion and conclusions were well described.

I only have a few small comments:

-     In abstract, the sentence in lines 19-20 is not clear. This sentence should be as follows: “Chickens were fed with corn-soybean meal diet supplementation with addition of 0, 150, 300 and 450 mg/kg XOS for 42 days.”

-          I would advise you to change the keywords so that they are not the same as in the title of the manuscript, possibly you can change the title of the manuscript a bit

-          The sentence in line 51 should start with a new paragraph. Also the sentence in line 68.

-          I missed a few words about feeding behavior in the Introduction

-          The subheading 2.1 in Materials and Methods is redundant

-          In line 87 should be: “The experiment trial period.....”

-          In line 91 should be the past tense: were instead of are

-          Explain the abbreviation GSH-Px in line 270

Conclusion:  The manuscript contributes to the knowledge within the scope of the journal. This paper should be considered as acceptable for publication in Animals, because this paper is a significant and important contribution to the field of investigation.

language is good and understandable

Author Response

Point 1: In abstract, the sentence in lines 19-20 is not clear. This sentence should be as follows: “Chickens were fed with corn-soybean meal diet supplementation with addition of 0, 150, 300 and 450 mg/kg XOS for 42 days.”

Response 1: I have revised it to "Chickens were fed with corn-soybean meal diet supplementation with addition of 0, 150, 300 and 450 mg/kg XOS for 42 days.” in line 21.

Point 2:  I would advise you to change the keywords so that they are not the same as in the title of the manuscript, possibly you can change the title of the manuscript a bit.

Response 2: I have changed the keywords in line 36: xylooligosaccharides; behavior; performance indicators; meat characteristics; broiler

Point 3: The sentence in line 51 should start with a new paragraph. Also the sentence in line 68.

Response 3: The sentence in line 51 and 68 have started with a new paragraph.

Point 4: I missed a few words about feeding behavior in the Introduction.

Response 4: I have supplemented with feeding behavior in the Introduction in lines 71-74: "Feeding behavior is a fundamental activity of all animals that can be modulated by internal energy states or external sensory signals, plays a decisive role in animal growth and reproduction, and promotes the expression of biological functions that are meaningful to animal welfare."

Point 5:  The subheading 2.1 in Materials and Methods is redundant.

Response 5: I have removed the subheading 2.1 in Materials and Methods.

Point 6:  In line 87 should be: “The experiment trial period.....”

Response 6: I have revised it to "The experiment trial period lasted for 42 days and divided into two stages" in line 91.

Point 7:  In line 91 should be the past tense: were instead of are

Response 7: I have revised it to "were" in line 95.

Point 8: Explain the abbreviation GSH-Px in line 270.

Response 8: The reference was deleted because the discussion section was changed.

Reviewer 2 Report

Dear authors,

The aim of the research should be highlighted in the summary.

In the results section, only the significance different of individual indicators between groups is highlighted. It is necessary to explain how supplementation XOS affects indicators that differ significantly between experimental groups.

In Table 8, explain the fact that the value of pH24 is higher than pH45 min in the muscle of the thigh.

On average, the addition of XOS significantly increases the pH45 values in the meat of thigh, how do you explain this?

Further clarify the pH values of the thigh where you have a lower pH45 (6.38) compared to pH24 (6.54) which is not usual. The pH value after slaughtering chickens and cooling the meat decreases.

In Table 8, for the Cooking loss (%) and Shere force (N) indicators for the L effect, the p value is statistically significant. This significance is not marked by letters a,b between groups.

In the same table, at the value of pH45, check the letter markings which are related to statistical significance.

Q: Quadratic effect.  In the table, please mark the p value column for this effect with Q instead of C.

Line 100-102 In the materials and methods, it is written that the research used 240 chickens divided into 4 groups (treatments), and each group had 6 repetitions with 10 chickens per repetition. Explain how you chose 8 chicks per treatment to monitor behavior, if you have 6 replicates per treatment.

Does this mean that in a particular repetition 2 birds in a group of 10 were tracked?

Line 269-275 Given that in this research you did not analyzed enzymatic activity (GSH-Px), it is necessary to further clarify the influence of XOS supplement on meat quality indicators. The current comment is only a guess, not a product of your measurements. Line 269-275 Given that in this research you did not analyzed enzymatic activity (GSH-Px), it is necessary to further clarify the influence of XOS supplement on meat quality indicators. The current comment is only a guess, not a product of your measurements.

The conclusion needs to be refined. It is necessary to mention the results of XOS influence on chicken meat quality indicators.

Author Response

Point 1: The aim of the research should be highlighted in the summary.

Response 1: I have revised it to “However, it is unclear whether feeding broiler chickens with different levels of xylooligosaccharides (XOS) affects the feeding behavior, thereby affecting broilers growth performance, and whether the addition of XOS has an effect on meat quality.” in lines 13-16.

Point 2: In the results section, only the significance different of individual indicators between groups is highlighted. It is necessary to explain how supplementation XOS affects indicators that differ significantly between experimental groups.

Response 2: I have revised the results section in lines 163-166, 169-174, 202, 209-214.

Point 3: In Table 8, explain the fact that the value of pH24 is higher than pH45 min in the muscle of the thigh. Further clarify the pH values of the thigh where you have a lower pH45 (6.38) compared to pH24 (6.54) which is not usual. The pH value after slaughtering chickens and cooling the meat decreases.

Response 3: We removed the abnormal value and re-analyzed the pH24h of thigh muscle, and the corrections are made in Table 8, pH24h value of thigh muscle in control group was 6.31.

Point 4: On average, the addition of XOS significantly increases the pH45 values in the meat of thigh, how do you explain this?

Response 4:  I have explained that in lines 283-286: "Research shows that XOS promotes the absorption of minerals by the body [46]. When the content of Mg2+ in blood increases, it is beneficial to increase the pH value of muscle, reduce the speed of glycolysis, and thus maintain the pH value of muscle at a high level."

Point 5: In Table 8, for the Cooking loss (%) and Shere force (N) indicators for the L effect, the p value is statistically significant. This significance is not marked by letters a,b between groups.

Response 5: I have marked the significance in Table 8.

Point 6: In the same table, at the value of pH45, check the letter markings which are related to statistical significance.

Response 6: I have revised the letter in Table 8, the value of pH45min in breast muscle was 6.66 a.

Point 7: Q: Quadratic effect.  In the table, please mark the p value column for this effect with Q instead of C.

Response 7: I have replaced the Cubic effect in each table to Quadratic effect, and marked the p value.

Point 8: Line 100-102 In the materials and methods, it is written that the research used 240 chickens divided into 4 groups (treatments), and each group had 6 repetitions with 10 chickens per repetition. Explain how you chose 8 chicks per treatment to monitor behavior, if you have 6 replicates per treatment. Does this mean that in a particular repetition 2 birds in a group of 10 were tracked?

Response 8: We selected 4 replicates from each group and 2 chickens from each replicate for behavioral observation.

Point 9: Line 269-275 Given that in this research you did not analyzed enzymatic activity (GSH-Px), it is necessary to further clarify the influence of XOS supplement on meat quality indicators. The current comment is only a guess, not a product of your measurements.

Response 9: I have supplemented with the influence of XOS on meat quality indicators in lines 273-276: "A previous study showed that the application of XOS in growing-finishing pigs can decrease the drip loss of dorsal longissimus muscle, indicating that XOS has the effect of improving muscle water-holding capacity, and this effect was gradually strengthened with the increase of XOS supplementation [43]."

in lines 283-286: "Research shows that XOS promotes the absorption of minerals by the body [46]. When the content of Mg2+ in blood increases, it is beneficial to increase the pH value of muscle, reduce the speed of glycolysis, and thus maintain the pH value of muscle at a high level. "

in lines 289-291: "XOS as an antioxidant can scavenge free radicals produced by cell metabolism, and alleviate the lipid peroxidation of cell membrane [49], this may be the reason for the decreased cooking loss of the muscle."

Point 10: The conclusion needs to be refined. It is necessary to mention the results of XOS influence on chicken meat quality indicators.

Response 10: I have refined the conclusion in lines 293-296: "this study confirmed that XOS can promote feeding behavior and produce positive effects on growth performance through increasing ADG during the grower stage. At the same time, dietary supplementation of XOS can also improve pH value and reduce cooking loss of meat after slaughter."

Reviewer 3 Report

The manuscript entitled "Feeding Behavior, Growth Performance and Meat Quality Profile in Broiler Chickens Fed Multiple Levels of Xylooligosaccharides" was reviewed.

Although the study provides good information regarding the use of increasing levels of XOS on eating behavior, the results and discussion do not contribute anything new on the subject since the authors do not scientifically relate their findings to the mechanisms of action of XOS.

For instance, there is not effort to explanin why or how the increased percentage of feeding time, longer average feeding time, and improved feeding frequency is driven by an increase in XOS (related to the mecanisms of action of XOS like the prebiotic properties). Nro is it explained the changes in feeding behavior with high XOS levels and reduced performance. 

In the Discussion also, the theories of hunger and satiety are exposed, but a connection between the changes in eating behavior with the use of XOS and these theories is not presented either. If broilers had ad libitum access to feed, would they experience hunger? 

For these reasons I am sorry to reject the manuscript. 

English need to be improved.

Round 2

Reviewer 3 Report

Unfortunatelly the question I raised were not answered that´s why I keep the same view to reject the manuscript 

The Eglish needs minor corrections 

Author Response

Point 1: There is not effort to explanin why or how the increased percentage of feeding time, longer average feeding time, and improved feeding frequency is driven by an increase in XOS (related to the mecanisms of action of XOS like the prebiotic properties). No is it explained the changes in feeding behavior with high XOS levels and reduced performance. 

Response 1: I have revised it to “Research shows that there was a significant negative correlation between the daily feeding frequency and the ADG of chickens, and breeding chickens with fewer feeding frequency could improve the production performance [35].” in lines 254-256.

“XOS can enhance the expression of genes involved in carbohydrate transport metabolism such as ABC transporter, amylase, xylosidase, galactosidase and glucosidase [40]. The percentage of feeding time had increased during 22–42 days, while the supplementation of 300 and 450 mg/kg XOS had no effect on growth performance. This possibly due to the overacidic intestinal environment affects the activity of digestive enzymes or XOS promote the rate of empting the digestive tract, and adversely affecting the absorption of nutrients.” in lines 265-272.

Point 2: In the Discussion also, the theories of hunger and satiety are exposed, but a connection between the changes in eating behavior with the use of XOS and these theories is not presented either. If broilers had ad libitum access to feed, would they experience hunger? 

Response 2: I have revised it to “A higher and longer satiation reduces the motivation to feed for a longer time and increases the intervals between meals [28]. On the contrary, the sense of hunger will stimulate the feeding behavior and improve feeding frequency. This may be related to the function of XOS in promoting intestinal peristalsis and accelerating intestinal emptying rate [29]. The effect of promoting intestinal tract movement was strengthened with the increase of XOS supplemental level, and feed was excreted before it was fully digested and absorbed. Therefore, broilers could meet their food requirements by increasing feeding time and frequency.” in lines 232-239.

Round 3

Reviewer 3 Report

I have no more comments 

It requires minor editing